# Cerium Oxide Enhances the Toxicity of Zinc Oxide Nanoparticles in Human Lung Epithelial Cell Cultures

**DOI:** 10.3390/toxics10090522

**Published:** 2022-09-01

**Authors:** Tasnim Al Rashaideh, Nervana Metwali, Sarah S. Perry, Andrea Adamcakova-Dodd, Peter S. Thorne

**Affiliations:** 1Human Toxicology Program, Graduate College, University of Iowa, Iowa City, IA 52242, USA; 2Department of Occupational and Environmental Health, College of Public Health, University of Iowa, Iowa City, IA 52242, USA

**Keywords:** cerium oxide, lung epithelial cells, nanotoxicology, nanoparticles, tissue culture, zinc oxide

## Abstract

Recently, many approaches have been developed to improve the performance of nanomaterials. Combining more than one nanomaterial is one such approach that achieves superior results. However, during the fabrication of nanomaterials or formulation of end products, materials can be released into the ambient air and be inhaled by workers. The adverse health outcomes of inhaling such compounds are unknown. In this study, we examined such effects in combining two of the most utilized nanomaterials in several industrial sectors: zinc oxide (ZnO) and cerium oxide (CeO_2_). These materials can be found together in sunscreens, polyvinyl alcohol (PVA) films, and construction products. The aim of this study was to assess the adverse biological outcomes of CeO_2_–ZnO nano-mixtures in human lung epithelial cells. A549 human lung epithelial cells were treated with increasing concentrations of ZnO or CeO_2_ NPs alone, or as a mixture of both, under submerged conditions for 24 h. After treatment, cell viability, reactive oxygen species (ROS) formation, cell membrane integrity, and cytokine production were examined. ZnO NPs showed a dose-dependent trend for all endpoints. CeO_2_ NPs did not exhibit any toxic effect in any individual concentrations. When higher doses of ZnO were combined with increasing doses of CeO_2_, loss of cell viability and an elevation in cell membrane leakage were observed. Interleukin 8 (IL-8) and ROS generation were higher when ZnO NPs were combined with CeO_2_ NPs, compared to cells that were treated with ZnO alone. The release of monocyte chemoattractant protein-1 (MCP-1) was reduced in the cells that were treated with higher doses of ZnO and CeO_2_. Thus, the presence of CeO_2_ enhanced the toxicity of ZnO in A549 cells at non-toxic levels of CeO_2_. This suggests an additive toxicity of these two nanomaterials.

## 1. Introduction

With the increased demand for new and superior materials in various nanotechnology applications; manufacturers and material scientists have begun to incorporate more than one nanomaterial into formulations [1,2]. For instance, tannin that has zinc oxide (ZnO) and cerium oxide (CeO_2_) nanoparticles (NPs) is considered to be a superior wood coating that protects against weathering and stabilizes color [3]. Research has shown that ZnO–CeO_2_ nanocomposites improve the structural and electrical properties of polyvinyl alcohol (PVA) films, which are used to manufacture the dissolvable packaging used in dishwashing cubes [4]. New generations of sunscreens have adopted ZnO and CeO_2_ NPs in their formulations, to replace NPs that are harmful to marine life [5]. Exposure to nanomaterials in different circumstances increases the concern about potential health implications. The European Risk Observatory of the European Agency for Safety and Health at Work categorizes NPs and ultrafine particles as being among the top 10 most hazardous chemicals in occupational settings [6]. An inhalation study of ZnO NPs on 16 human subjects found that inhaling 1 mg/m^3^ or more of ZnO for up to 4 h caused significant increases in blood parameters such as C-reactive protein and serum amyloid A [7]. In animal studies, ZnO caused lung inflammation, enhanced the release of lactate dehydrogenase (LDH), and affected cell cycle genes and DNA repair pathways [8,9].

In vitro studies found that ZnO induced damage at the proteomic and genomic level in A549 cells, through lipid peroxidation, reduction in GSH levels, and DNA damage [10]. Both in vitro and in vivo studies agreed that ZnO NPs cause toxic effects at the cellular and molecular level. However, the toxicity of CeO_2_ NPs is a controversial topic [11,12]. Many studies have shown that CeO_2_ NPs are non-toxic, while others have concluded the opposite [12,13].

The practice of incorporating more than one nanomaterial in the same environment can expose workers and end-product users to combinations of nanomaterials that have not been assessed for their combined toxicity. Potential toxic effects of exposure to various nanoparticle mixtures on human health are not known; thus, a toxicity assessment of individual and combined exposures to nanomaterials is indispensable.

Using an established in vitro system, the goal of this study was to assess the toxicity of ZnO and CeO_2_ when combined and compare this with their individual forms. We hypothesized that the presence of CeO_2_ NPs with ZnO NPs will augment their individual adverse biological outcomes in human lung epithelial cells. In order to achieve this, serial dilution for each of the NPs was performed to achieve a dose-response curve (Appendix A). The mixtures of CeO_2_ NPs with ZnO were prepared by sequentially combining each of the six dilutions of CeO_2_ with the four highest concentrations of ZnO. The NP exposure doses were selected based on our previous studies [14]. After treating cells for 24 h in submerged conditions with individual NPs or the mixtures, the cell viability, membrane integrity, generation of ROS, release of cytokines, and cell morphology were evaluated. Our findings showed that combining non-toxic doses of CeO_2_ with ZnO enhanced the toxic effects of ZnO NPs.

## 2. Materials and Methods

### 2.1. Characterization and Preparation of Nanomaterials Suspensions

The cerium oxide used was a 99.9% pure nano-powder with stated size range of 15–30 nm, (product ID 1406RE, Nanostructured & Amorphous Materials, Inc. Houston, TX, USA). Zinc oxide NPs had a manufacturer reported size of 10 nm (Meliorum Technologies, Inc. Rochester, NY, USA). The hydrodynamic diameter of agglomerates and zeta potential of nanomaterials suspended in ultrapure water were measured with a Zetasizer Nano-ZS (Malvern Instruments, Westborough, MA, USA). The size and shape of NPs were viewed with transmission electron microscopy imaging (TEM, EOL JEM-1230 Transmission Electron Microscope, Jeol, Japan). X-Ray photoelectron spectroscopy (XPS, Kratos Axis Ultra, Chestnut Ridge, NY, USA) was used to examine the oxidation status. Endotoxin levels were assessed using the kinetic chromogenic *Limulus* amebocyte lysate assay (Lonza Walkersville, Inc., Walkersville, MD, USA), as previously described [15]. Physicochemical properties of ZnO NPs were reported previously [16].

NP stocks of 500 µg/mL were freshly prepared before treatments, as described by Deloid et al. [17]. The stocks were vortexed for 30 s, then ultrasonicated in a cup-horn ultrasonic bath (Model # 431C2, Qsonica, CT) at 75% amplitude for specific times necessary to reach the critical delivered sonication energy (DSE_cr_), which was determined following a published protocol [17,18]. ZnO NPs required 6 min of sonication, and CeO_2_ NPs required 18 min. NPs stocks were further diluted with phenol red-free RPMI 1640 cell culture medium (Roswell Park Memorial Institute 1640, Life Technologies Corporation, Grand Island, NY, USA) supplemented with 10% fetal bovine serum (FBS, HyClone Laboratories, Inc., Logan, UT, USA) and 100 mg/mL penicillin streptomycin (Life Technologies Corporation, Grand Island, NY, USA). HEPES buffer (4-(2-hydroxyethyl) piperazineethanesulfonic acid, Life Technologies Corporation, Grand Island, NY, USA) was used to control the pH of the medium. Six desired concentrations of 2.0, 3.9, 7.8, 15.6, 31.3, and 62.5 µg/mL were prepared by two-fold dilutions of the NP stocks. These concentrations corresponded to 0.5, 1, 2, 4, 8, and 16 µg/cm^2^ per each NP per each well.

### 2.2. Cell Culture and Treatment

A549 Type II human pulmonary epithelial cells, passages 3–16 (American Type Culture Collection, #CCL-185, Manassas, VA, USA) were used in this study. The cells were cultivated in T-75 flasks with DMEM complete medium (Dulbecco’s Modified Eagle’s Medium, Life Technologies Corporation, Grand Island, NY, USA) supplemented with 10% fetal bovine serum, 100 mg/mL penicillin and streptomycin, and 25 mM of HEPES in 5% CO_2_, at 95% humidity and 37 °C. At 80% confluence, cells were harvested using 0.25% trypsin-EDTA (Sigma-Aldrich, St. Louis, MO, USA) and suspended in phenol red-free cell culture medium, then seeded in 12-well cell culture plates (Corning. Ref # 3515, with cell growth surface area of 3.8 cm^2^, Corning, NY, USA) at a density of 1.0 × 10^5^ cells per well in 2 mL of supplemented DMEM for 48 h. On the day of treatment, DMEM was pipetted off the cells, and they were washed with 1 mL of 1640 RPMI phenol red-free medium. Each treatment experiment consisted of 4 sets of 12-well plates; each well was designated for 1 dose, and 12 randomly selected wells were designated as controls. Cells were exposed to the NPs for 24 h. Five treatments were performed for viability assays and six for ROS assays. The remaining supernatant of each treatment was collected, centrifuged at 14,000× *g* for 30 min, and stored at −80 °C for cytokine and chemokine assays. Switching from DMEM to 1640 RPMI did not affect the cells’ doubling time or morphology. Initially, we were culturing and splitting the cells with DMEM, as recommended by the supplier. We used 1640 RPMI in the treatments since most researchers have used this medium for submerged nanotoxicology studies.

### 2.3. Viability Assay

AlamarBlue^®^ (Invitrogen, Eugene, OR, USA) assay was used, following the manufacturer’s instructions, to examine the viability of A549 cells after treatment with NPs. The resazurin-based reagent (blue) is reduced to resorufin in live cells and appears red-pink, whereas it remains blue in dead cells. After treatment of the cells with NPs, the reagent was diluted with 10% of supplemented phenol red-ree RPMI 1640, then 0.5 mL was added to each well. Cells were incubated for 4 h in standard cell culture conditions. After 4 h of incubation, 200 µL of samples were transferred to 96-well plates, and a microplate reader (SpectraMax M5, Molecular Device, San Jose, CA, USA) was used to detect fluorescence (excitation: 560 nm; emission: 590 nm). The viability of treated cells was calculated as follows, based on relative fluorescence unit (RFU) values of treated and control cells:(1)$$\%\;\mathrm{Viability}\;\mathrm{of}\;\mathrm{treated}\;\mathrm{cells}=\left(\frac{\mathrm{RFU}\;\mathrm{of}\;\mathrm{treated}\;\mathrm{cells}}{\mathrm{RFU}\;\mathrm{of}\;\mathrm{control}}\right)*100$$

The half-maximal inhibitory concentration (IC50) was calculated by plotting increasing concentrations of ZnO vs. the viability of A549 cells (dose-response data).
Y = a ∗ X + b, IC50 = (0.5 − b)/a.

The x-axis (dose) was logarithm-transformed.

### 2.4. LDH Leakage

Lactate dehydrogenase (LDH) leakage is an indicator of cell membrane damage. An LDH kit was used as recommended by the manufacturer (Roche Diagnostics, Penzberg, Germany) on cell culture supernatants to measure leakage of LDH. Standards (L-LDH from rabbit muscle, Roche Diagnostics) were run in duplicate, and each sample was run at three dilutions (1:2, 1:4, and 1:8) to test for assay interference by the NPs. Values reported are those at the dilution for which the assay fell within the detectable range, with no evidence of interference.

After adding the reaction components and incubating the plate in the dark at room temperature for 10 min, absorbance was measured at 450 nm using a microplate reader (SpectraMax M5). The LDH leakage of treated cells was calculated as follows, based on the absorbance values of treated and control cells:(2)$$\%\;\mathrm{LDH}\;\mathrm{leakage}\;\mathrm{of}\;\mathrm{treated}\;\mathrm{cells}=\left(\frac{\mathrm{absorbance}\;\mathrm{at}\;450\;\mathrm{nm}\;\mathrm{of}\;\mathrm{treated}\;\mathrm{cells}}{\mathrm{absorbance}\;\mathrm{at}\;450\;\mathrm{nm}\;\mathrm{of}\;\mathrm{control}}\right)*100$$

### 2.5. Reactive Oxygen Species

Reactive oxygen species (ROS) were detected using carboxy- 2′,7′-dichlorodihydrofluorescein diacetate (carboxy-H2DCFDA) (Invitrogen, 25 mg, catalog # C-400). After the treatment, cells were washed with phosphate buffer saline (PBS). The exposure medium was collected for further assays. Hanks Balanced Salt Solution (HBSS) containing carboxy-H2DCFDA (25 µM, final concentration) was added to the cells for 30 min of incubation at 37 °C under dark conditions. Following incubation, the HBSS solution was removed and the cells were washed twice with 200 uL of PBS; they were then lysed with 400 µL of 0.1% Triton X-100 in PBS for 30 min under dark conditions. After lysis, the suspensions were centrifugated at 14,000× *g* for 15 min at 4 °C, and the supernatant was taken and placed in a 96-well plate for measurement. The oxidation of the substrate carboxy-H2DCFDA by ROS produced a green-fluorescent emission. Fluorescence intensity in cells was measured using a microplate reader (SpectraMax M5) at excitation: 485 nm and emission: 530 nm.

### 2.6. Cytokines and Chemokines

The cytokine releases of IL-8, MCP-1, and IL-1β were measured in cell culture supernatant by ELISA, following the manufacturer’s protocol (BD Biosciences, San Diego, CA, USA and R&D, Minneapolis, MN, USA) in EIA/RIA polystyrene high binding plates (Corning, NY, USA. Ref # 2592). The plates were coated and refrigerated for 24 h, then used as described in the protocols. Standards were run in duplicate, and each sample was run at three dilutions (1:4, 1:8, and 1:16) in order to test for assay interference by the NPs. Values reported are those at the dilution for which the assay fell within the detectable range, with no evidence of interference.

### 2.7. Interference of NPs with Cytotoxic Assays

Nanoparticles can interfere with cytotoxicity assays in several ways, either by adsorbing to the assay components or affecting the optical measurements of the assay [19]. Interference of NPs with endpoints was assessed by centrifuging the suspension of NPs at 14,000× *g* for 30 min, after which the supernatant was used for LDH and cytokines assays. For viability and ROS, suspended NPs or NP pellets were mixed with assay reagents to test for assay interference.

### 2.8. Data Analysis and Visualization

All toxicity assay data were calculated as percent of controls. For LDH leakage, generation of ROS, IL-8, and MCP-1 cytokines/chemokines, the results were divided by the corresponding viability and expressed as mean ± SEM. Statistical analyses were performed using unbalanced two-way analysis of variance testing (SAS Version 9.4, SAS Institute, Cary, NC, USA) Bonferroni correction for multiple comparisons was used by multiplying the *p*-values with the number of comparisons. Treatments of cells were performed at least 11 times, 5 for viability assay and 6 for oxidative stress. LDH leakage was measured on the collected supernatant of 7 treatments, while MCP-1 and IL-8 were measured on 6 treatments.

## 3. Results

### 3.1. Characterization of Cerium Oxide Nanoparticles

TEM analysis showed that CeO_2_ NPs were spherical in shape and confirmed the manufacturer’s reported primary size range of 15–30 nm. Endotoxin was below the lower limit of detection (LOD = 0.024 EU/mL). Zeta-potential was −15 mV, and the hydrodynamic size was 445 ± 106 nm for the suspension of CeO_2_ NPs in ultrapure water (pH 8.4). The XPS spectrum of CeO_2_ showed O s1, and Ce 3d (Figure 1A). The Ce 3d region contained two parts: Ce 3d5/2 and Ce 3d3/2 (Figure 1B). The XPS spectrum of CeO_2_ NPs showed that the surface of the material contained both Ce^4+^ and Ce^3+^ oxidation states. Both oxides showed significant spin orbit coupling, splitting the 3d Ce (IV) peak into six peaks total, while the 3d Ce (III) peak split into four peaks total, for a total of ten peaks. Due to peaks overlapping, determining the dominant oxide form was challenging. 

### 3.2. Viability Assay

In order to understand the dose-response relationship, we identified the half-maximal inhibitory concentration (IC_50_) through the viability assay (AlamarBlue^®^) for each of the NPs. For ZnO, the IC_50_ was 21.4 µg/mL (Figure 2). For CeO_2_, the IC_50_ exceeded 500 µg/mL; none of the doses up to 500 µg/mL caused any measurable loss in viability.

ZnO NPs caused significant loss in A549 cell viability in a dose-dependent manner (Figure 3). When the dose of 15.6 μg/mL of ZnO was combined individually with all six doses of CeO_2_, the loss in viability at the highest concentration of CeO_2_ was more than the 15.6 μg/mL of ZnO alone. The mixture of 31.3 μg/mL of ZnO and 62.5 μg/mL of CeO_2_ caused the viability to be significantly lower than the 31.3 μg/mL of ZnO alone. When comparing 62.5 μg/mL of ZnO alone to 62.5 μg/mL of ZnO plus 31.3 μg/mL or 62.5 μg/mL of CeO_2_, there were no significant differences; however, a loss in viability was notable (Figure 3).

### 3.3. LDH Assay

CeO_2_ NPs did not affect the leakage of LDH at any dose. On the other hand, ZnO caused the elevated release of LDH in a dose-dependent manner. When 31.3 μg/mL of ZnO was combined with CeO_2_, all six doses (7.8–62.5 μg/mL) of CeO_2_ showed significant elevations in LDH release. When 62.5 μg/mL of ZnO alone was combined with increasing doses of CeO_2_ (7.8–62.5 μg/mL), the LDH release was significantly higher than 62.5 μg/mL of ZnO alone (Figure 4).

### 3.4. ROS

The potential generation of intracellular ROS was detected using carboxy-H_2_DCF-DA. For cells treated with CeO_2_, oxidation of the probe was no different from that of the control. However, ZnO showed an increase in oxidation of the probe in a dose-dependent manner. Mixing CeO_2_ with ZnO increased the oxidation of the probe in all samples. A significant increase in ROS was notable after the addition of 62.5 μg/mL of CeO_2_ to 7.8 or 62.5 μg/mL of ZnO. These results suggest an increase in ROS (Figure 5).

### 3.5. Cytokines

CeO_2_ (all doses) did not affect the release of IL-8 from the cells; however, ZnO enhanced the release of IL-8 in a dose-dependent manner. When 62.5 μg/mL of ZnO alone was combined with increasing doses of CeO_2_, the release of IL-8 was higher than 62.5 μg/mL of ZnO alone. At 62.5 μg/mL of CeO_2_, the release of IL-8 was significantly higher than 62.5 μg/mL of ZnO alone (Figure 6). The same trend was noted when 31.3 μg/mL of ZnO was mixed with increasing doses of CeO_2_ NPs. For MCP-1, increasing the doses of ZnO affected MCP-1 release only at 62.5 μg/mL. The mixture of 62.5 μg/mL of ZnO and 62.5 μg/mL of CeO_2_ caused a significant reduction in MCP-1 (Figure 7). IL-1β was not detectable in either treated or control cells at the 24-hour post-exposure time point. 

## 4. Discussion

Studies have used various models to evaluate the toxicity of monotypic nanoparticles (NPs). Studies on NP mixtures have primarily focused on other biological systems, such as aquatic organisms and bacteria as a result of the serious environmental complications these mixtures may cause [20,21,22]. Studies on how combined nanomaterials affect human health are extremely limited, and no inhalation dose-response studies have been performed using nano-mixtures. The present study utilized a submerged in vitro model and human lung cell line as a screening method. In vitro submerged toxicity studies are used as the first step to assess the biological consequences that result from NP exposure [23]. Historically, most in vitro studies have utilized commercial cell lines rather than primary cells. The main reasons for using commercial cell lines over primary cells include their high reproducibility, immortality, comparability, usability, and lower cost [24].

In the present study, using an in vitro submerged cell culture to combine ZnO NPs with CeO_2_ NPs showed enhanced toxicity over the effect of individual NPs. The cytotoxicity of the NPs was examined by measuring the viability of the cells, the elevation in lactate dehydrogenase (LDH), the generation of ROS, and the release of chemokines/cytokines. Individual CeO_2_ NPs did not impact any of the examined endpoints. This finding is in line with several previous studies that found CeO_2_ NPs cause no toxic effects at environmentally-relevant doses [12,14]. Conversely, some studies have found that CeO_2_ leads to a loss in cell viability [25]. One study explained this controversy around CeO_2_ NP toxicity by elucidating that the oxidation state of Ce NPs affects the toxicity of the nanomaterial. Nanomaterials with a higher proportion of Ce^3+^ have been shown to be less toxic than those with a higher Ce^4+^ content [12]. The XPS spectrum of the CeO_2_ NPs used in the current study revealed that both oxidation states of Ce existed in the material we used. It is likely that Ce^3+^ was dominant over Ce^4+^ since no toxic effect was noticed. It is also worth noting that ZnO NPs caused cytotoxic effects in a dose-dependent manner, as shown by others [14,26]. Overall, the combination of CeO_2_ and ZnO augmented the toxic effects of ZnO NPs and caused a reduction in cell viability. This finding was confirmed by the LDH data. Microscopic images supported the results of the cell viability assay and the LDH assay. Similar findings were reported in another in vitro study that used immortalized T-lymphocytes to demonstrate that the presence of both CeO_2_ and ZnO NPs in the same environment reduced cell viability [27].

The mechanism behind this augmented toxicity is unclear. One possible explanation is that the oxidation of Ce^3+^ increases, causing the Ce^4+^ form to become dominant. This theory could be confirmed by closely examining the XPS spectrum of CeO_2_ and dissecting the overlapped peaks after preparing the mixtures of CeO_2_ and ZnO. Another explanation is that the dissolution of ZnO increases in the presence of CeO_2_; however, this was not investigated in this study. One could argue that at higher concentrations, NPs may precipitate to a greater extent. In order to evaluate this possibility, we treated the cells with a concentration of 500 μg/mL of CeO_2_ for 24 h. In turn, no loss in viability was observed. Although a concentration as high as 500 μg/mL of CeO_2_ NPs is unrealistic for inhalation studies [28], it was tested to confirm that the issue was not related to the suspension’s concentration but associated with the presence of two distinct nanomaterials in the same medium. In a separate publication, we performed dissolution studies to evaluate the solubility of ZnO NPs [16]. These studies showed that 100% of ZnO dissolved within 24 h in artificial lysosomal fluid (ALF) with pH 4.5 but <1% in Gamble’s solution (pH 7.4) after 2 weeks. Based on these results, in media with a pH near 7.4, we expect there would be little dissolution; however, ZnO NPs would likely dissolve intracellularly, as shown by their dissolution in ALF. Nevertheless, this is a very simplified assumption that is based mainly on pH; other parameters such as the presence of chelating agents in the media, agglomeration state of NPs, and dissolved organic matter plays a role in NP dissolution [29,30], and would need to be investigated further. Even though an in vitro study that investigated dissolution of ZnO NPs in DMEM [31] in a different type of cell (human epidermal cells), only dissolution of ZnO NPs with longer incubation times (over 24 h) was observed, attributing its toxicity to ZnO NPs entering the cells rather than ions.

The generation of ROS is a common mode of toxicity for several NPs, including ZnO NPs [8,10,32,33,34]. The ZnO NPs that were used in this study caused an increase in ROS that is similar to what other studies found [13,35]. As previously reported, CeO_2_ did not result in an increase in ROS generation [13]. For cells treated with CeO_2_ and ZnO NPs together, a significant increase in ROS was noted after adding 62.5 μg/mL of CeO_2_ to 7.8 μg/mL of ZnO and 62.5 μg/mL of ZnO. These results suggest that CeO_2_ enhances the role of ZnO in generating ROS. Nonetheless, many studies have suggested that CeO_2_ NPs act as antioxidants and scavengers for free radicals [36,37]. This antioxidant activity is supposed to reduce the generated ROS. Surprisingly, the presence of CeO_2_ along with ZnO enhanced the generation of ROS.

Cytokines and chemokines are small proteins produced by multiple cells, including lung cells, in response to several types of stress. In the current study, treating A549 cells with ZnO elevated IL-8 in a dose-dependent manner. This finding is in line with the results of a previous study [38]. The secretion of IL-8 was elevated similarly as in other investigated endpoints when ZnO was added to the CeO_2_ NPs. On the other hand, MCP-1 secretion was only noticed at the highest dose of ZnO NPs. A significant decrease in the release of MCP-1 mediator occurred when CeO_2_ was added. IL-1β is known to be produced by A549 cells [39]. In the present study, IL-1β was not detectable in either the treated or control cells. It may be that activation of the IL-1β pathway and cleavage of IL-1β did not occur within these cells to illicit IL-1β release. This finding could also be explained by the mediator’s short lifetime [39,40,41].

Morphological changes in cells after exposure to NPs in A549 cells have been reported previously [42]. In the present study, an increased number of dead cells was identified in the microscopic images of those cells after the treatment. These cells were bright and had detached from the wells to form clumps (Appendix A). Spicules were detected in the surviving cells. Many of the morphological changes we observed were also noted in a previous study; Santimano et al. found that A549 cells lost cell–cell contact after treatment with ZnO NPs [42]. Although A549 cells do not form tight junctions, the emptier areas were likely due to the detachment of cells.

One of the challenges in nanotoxicity assessment studies is the lack of a standardized methodology. Deloid et al. created a standardized protocol to prepare NPs for these types of studies [17]. According to that protocol, since every instrument has a specific frequency and amplitude and delivers a certain amount of energy, the ultrasonication apparatus must be calibrated through a calorimetric approach before manipulating the nanomaterials [17,43] (Appendix A). Determining the optimal duration of sonication for NPs is critical to establishing a stable suspension with the lowest possible hydrodynamic diameter [17]. In the current study, we saw that excessive sonication can induce the NPs to re-agglomerate (Appendix A). This pattern was in line with another calibration study that demonstrated that an increase in sonication time leads to re-agglomeration [43]. The study proposed that the increase in the suspension temperature enhanced the motion of particles in water and caused the particles to re-agglomerate. The cup-horn ultrasonicator was favored over the probe sonicator because the latter releases metal particles from the probe into the suspension, leading to contamination [17,44]. Together, the zeta potential, TEM image, and hydrodynamic diameter all confirm that CeO_2_ has a high tendency to agglomerate. This may explain the longer sonication time for CeO_2_ (18 min) than ZnO (6 min) to reach the lowest possible hydrodynamic size [16]. As recommended in the Deloid et al. protocol, the concentration of the stock for the preparation of NPs suspensions was 500 µg/mL [17]. To reduce the pipetting error, a two-fold serial dilution was used to prepare the treatment doses (Appendix A).

Although the in vitro submerged method, which was used to investigate the toxicity of the mixture, is usually the first approach utilized in in vitro toxicity assessment, it has several limitations. Its main limitation in this study is that the in vitro submerged approach could not represent the actual physiological conditions of the respiratory system [45]. Another limitation was the use of one type of commercial cell line rather than a variety of cell lines or primary cells to create a more realistic environment. The deposition of NPs in this method does not represent the actual manner in which NPs are deposited deep in the alveoli. An in vivo study, along with a co-culture study at an air–liquid interface, may provide a better understanding of the biological consequences of inhaling ZnO and CeO_2_ nano-mixtures.

Combining CeO_2_ with ZnO NPs augmented the toxic effects of ZnO. As such, this study may contribute to establishing guidelines to protect workers from NP mixtures and inform the design of safer products that use nanomaterials. There is a growing list of NPs that are being used in combination, and there are situations where exposure to several NPs occurs in humans and in ecosystems.

There are several limitations of our study. We have investigated only two types of NPs and their effects on one cell type. However, our previous studies that compared the toxicity of NPs in A549 cells and human primary bronchial epithelial cells (HBEC) [46] found similar results between the two cell types for viability, LDH release, and IL-8 production. HBECs derived from cadaver lungs have the downside of an unknown exposure history to, e.g., cigarette smoke, vaping, or occupational exposures. The uncertainty of cell dosimetry is another limitation of this study. The distorted grid (DG) computational model has been suggested to predict the dosimetry of certain nanosuspensions [17]. This model was not applied in this study due to the absence of characterization parameters, such as the average effective density of particle agglomerates for the 36 different individual and mixture concentrations studied. Our future research will more fully characterize the mixtures and attempt to validate the DG model for NP combinations, utilizing more advanced biological models and an expanded list of endpoints. Lastly, a more expansive physicochemical characterization of the nanomaterials in media would have aided interpretation of our results.

## 5. Conclusions

The presence of several types of nanomaterials together may affect their activity and toxicity. CeO_2_ is known to be non-toxic and did not cause any effect on any of the examined endpoints. However, when present with ZnO, CeO_2_ augmented the toxic effects of ZnO NPs. The viability of lung cells, the generation of ROS, the release of LDH, and the release of cytokines and chemokines were the primary endpoints of this study. Other endpoints, such as DNA damage, lipid peroxidation, and release of other inflammatory mediators, are also worth investigating.

With all the challenges that the in vitro submerged approach has, it is still a useful method to employ in toxicity screening. This method can be used effectively to investigate a large number of nano-combinations, before moving toward more complex models.

## Figures and Tables

**Figure 1 toxics-10-00522-f001:**
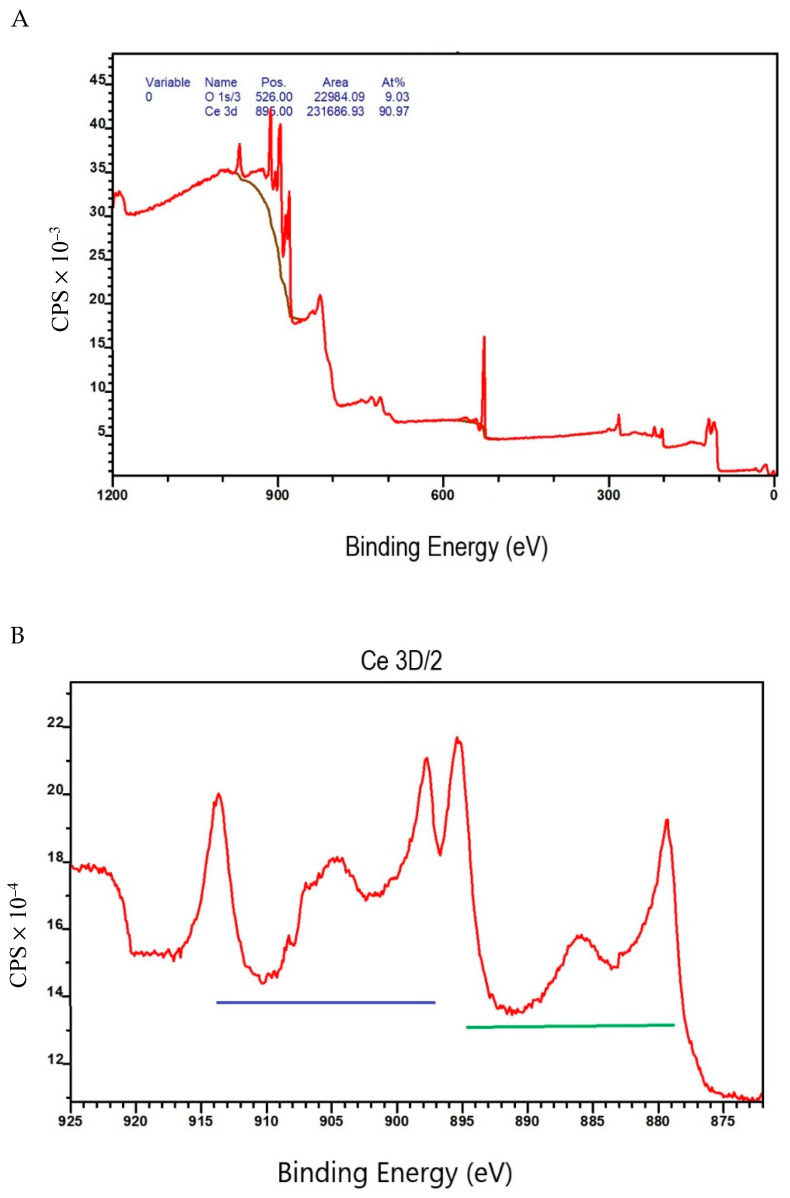
CeO_2_ XPS survey. (**A**) Full spectrum of CeO_2_ containing O s1 and Ce 3d; (**B**) detailed survey of Ce 3 d/2 region. CPS = counts per second.

**Figure 2 toxics-10-00522-f002:**
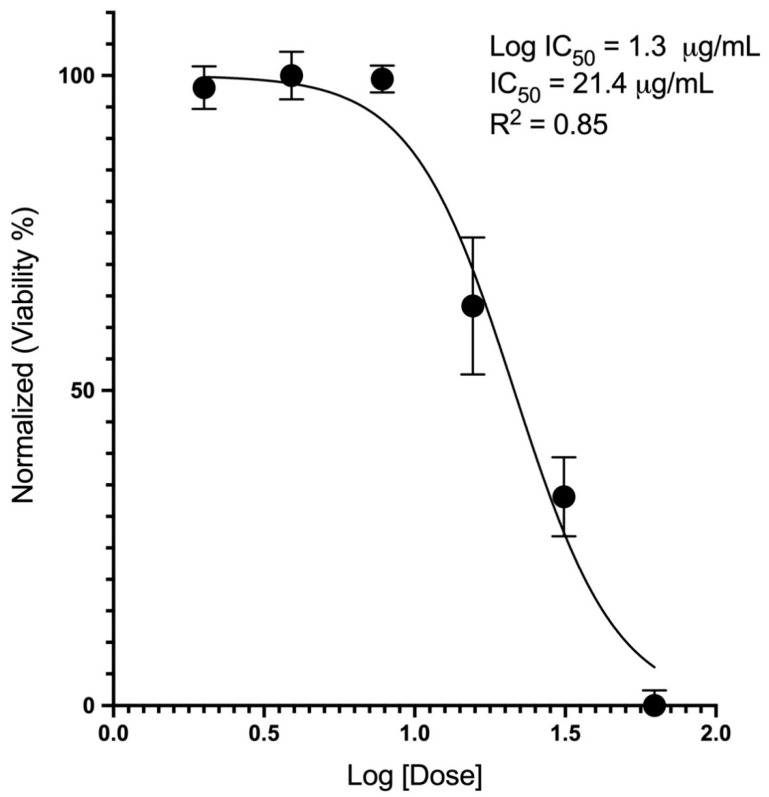
Dose-response curve of ZnO NPs after treating A549 cells for 24 h. ZnO showed a loss in viability of the cells in a dose-dependent manner. N = 8 for each treatment.

**Figure 3 toxics-10-00522-f003:**
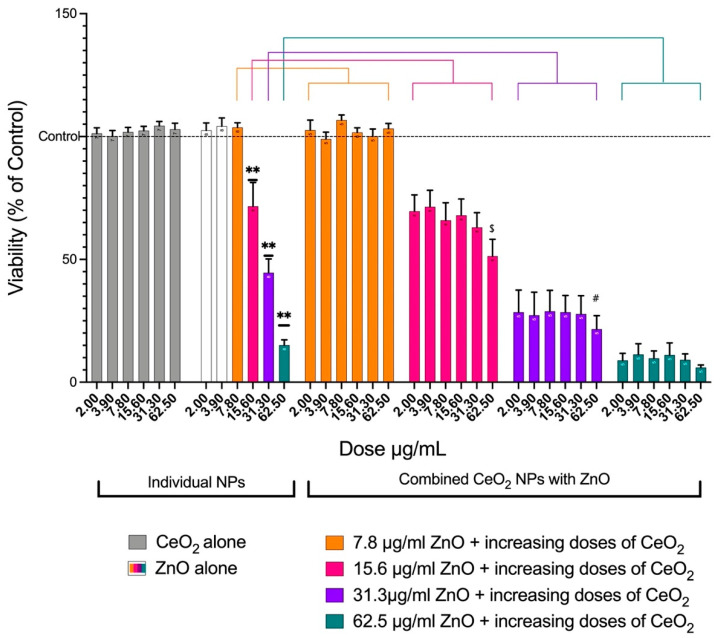
Viability of A549 cells after treatment with either individual NPs (two groups) or combined NPs (four groups). The plot shows the mean ± SEM of the percent AlamarBlue reduction in A549 cells (n = 5). For individual NPs, CeO_2_ alone showed no loss in viability, whereas ZnO showed a dose-dependent loss in viability. For the combined groups, combining each of the six doses of CeO_2_ with 7.8 μg/mL of ZnO did not affect the viability (orange). When each of the six doses of CeO_2_ was combined with 15.6 μg/mL of ZnO, the loss in viability at the highest concentration of CeO_2_ was more than that when 15.6 μg/mL of ZnO alone was used (pink). When 31.3 μg/mL of ZnO was added to the increasing doses of CeO_2_, 62.5 μg/mL of CeO_2_ caused the viability to be significantly lower than when 31.3 μg/mL of ZnO alone was used (purple). When 62.5 μg/mL of ZnO alone was compared with 62.5 μg/mL of ZnO plus 31.3 μg/mL or 62.5 μg/mL of CeO_2_, no significant difference was observed, but a loss in viability was notable. (**) Significantly different (*p* < 0.01) from the control. ($) Significantly different (*p* < 0.05) from 15.6 μg/mL of ZnO alone. (#) Significantly different (*p* < 0.05) from 31.3 μg/mL of ZnO alone.

**Figure 4 toxics-10-00522-f004:**
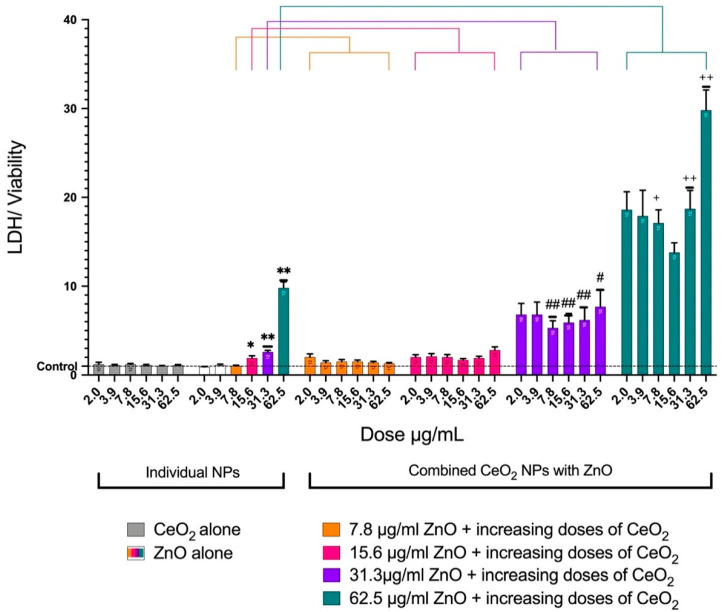
Release of lactate dehydrogenase from cells after treatment with either individual NPs (two groups) or combined NPs (for groups) for a 24-hour treatment. The graph represents the mean ± SEM of at least 10 treatments. The LDH was normalized for cell viability, LDH (% of control)/viability (% of control). The CeO_2_ group (gray) had no effect on cell viability. Increasing the doses of ZnO only caused an increase in LDH release in a dose-dependent manner. The first two groups of the combined NPs (orange and pink) did not show any increases in LDH leakage. When 31.3 μg/mL of ZnO was combined with CeO_2_, a significant elevation in LDH release was observed (purple). When 62.5 μg/mL of ZnO was combined with increasing doses of CeO_2_, the LDH release was significantly higher than when 62.5 μg/mL of ZnO alone was used (teal). (*) Significantly different (p < 0.05) from the control. (**) Significantly different (*p* < 0.01) from the control. (#) Significantly different (*p* < 0.05) from 31.3 μg/mL of ZnO alone. (##) Significantly different (*p* < 0.01) from 31.3 μg/mL of ZnO alone. (+) Significantly different (*p* < 0.05) from 62.5 μg/mL of ZnO alone. (++) Significantly different (*p* < 0.01) from 62.5 μg/mL of ZnO alone.

**Figure 5 toxics-10-00522-f005:**
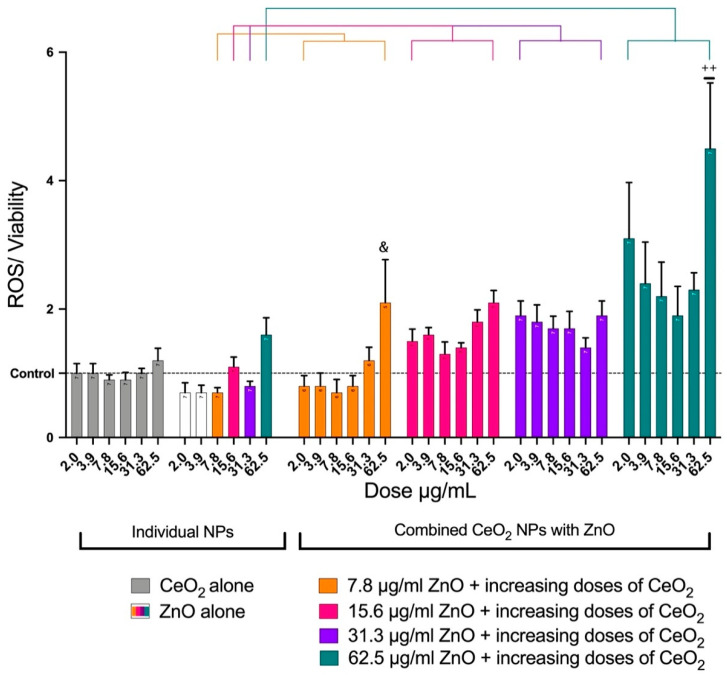
Generation of ROS after treating A549 cells with NPs for 24 h. The plot shows the mean ± SEM of ROS generation (n = 7) in A549 cells treated with ZnO, CeO_2_, or a mixture of both. ROS was normalized for cell viability, ROS (% of control)/viability (% of control). The CeO_2_ group (gray) had no effect on ROS. Increasing doses of ZnO only caused an increase in ROS generation. The mixture of both NPs caused more toxicity than ZnO alone. The presence of CeO_2_ augmented ZnO toxicity. (&) Significantly different (*p* < 0.05) from 7.8 μg/mL of ZnO alone. (++) Significantly different (*p* < 0.01) from 62.5 μg/mL of ZnO alone.

**Figure 6 toxics-10-00522-f006:**
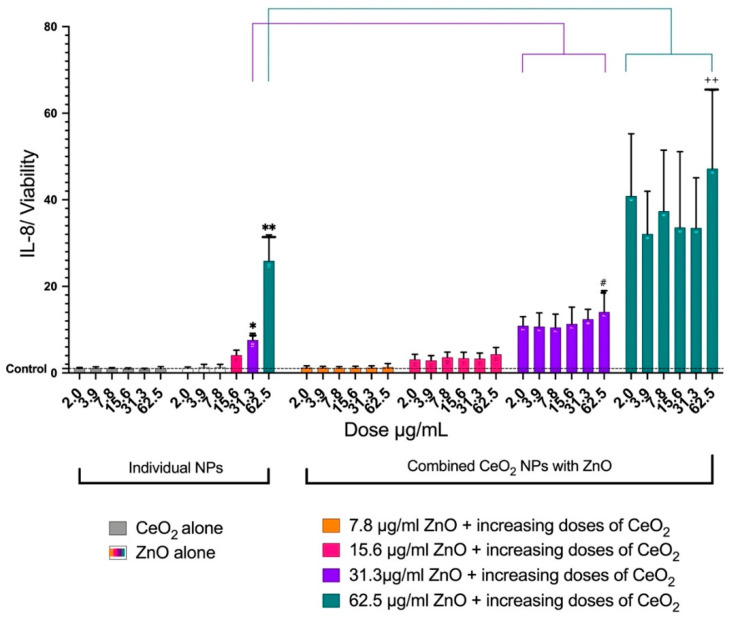
The effects of ZnO and CeO_2_ NPs in individual and mixture forms on the release of IL-8 from A549 cells. The plot shows the mean ± SEM of IL-8 release of six treatments of A549 with ZnO, CeO_2_, or mixtures of both. Levels of IL-8 were normalized for cell viability, IL-8 (% of control)/viability (% of control). The CeO_2_ group (gray) had no effect on IL-8 release. Increasing doses of ZnO only caused an increase in IL-8 release. Mixtures of both NPs caused more toxicity than ZnO alone. The presence of CeO_2_ augmented ZnO toxicity. (*) Significantly different (*p* < 0.05) from the control. (**) Significantly different (*p* < 0.01) from the control. (#) Significantly different (*p* < 0.05) from 31.3 μg/mL of ZnO alone. (++) Significantly different (*p* < 0.01) from 62.5 μg/mL of ZnO alone.

**Figure 7 toxics-10-00522-f007:**
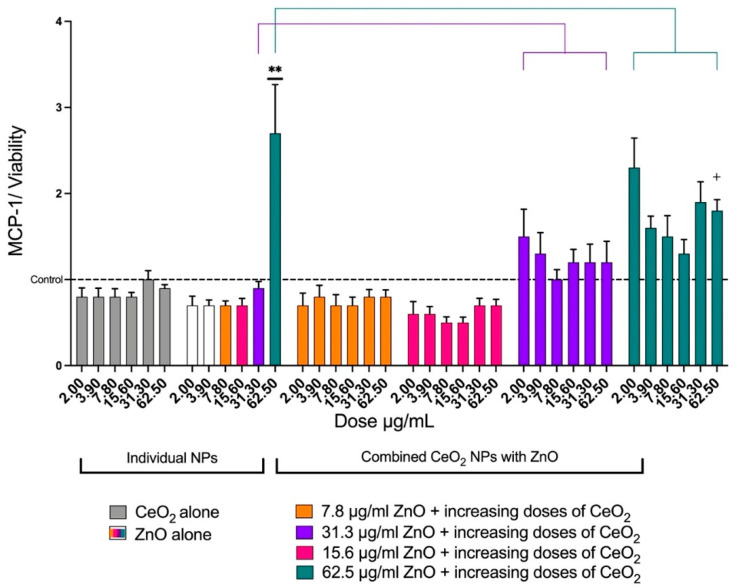
The effects of ZnO and CeO_2_ NPs in individual and mixture forms on the release of MCP-1 from A549 cells. The graph presents the mean ± SEM of MCP-1 release from six treatments of A549 cells with ZnO, CeO_2_, or mixtures of both. Levels of MCP-1 were normalized for cell viability, MCP-1 (% of control)/viability (% of control). CeO_2_ did not cause an elevation in MCP-1 release (gray). Increasing the doses of ZnO affected MCP-1 release only at 62.5 μg/mL. The mixture of 62.5 μg/mL of ZnO and 62.5 μg/mL of CeO_2_ caused a significant reduction in MCP-1 (teal). (**) Significantly different (*p* < 0.01) from the control. (+) Significantly different (*p* < 0.05) from 62.5 μg/mL of ZnO alone.

## Data Availability

The data sets generated and used during the current study are available from the corresponding author on reasonable request.

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
