# Peer review of "Cerium Oxide Enhances the Toxicity of Zinc Oxide Nanoparticles in Human Lung Epithelial Cell Cultures"

_toxics, 2022, doi:10.3390/toxics10090522_

Round 1

Reviewer 1 Report

Review of article Toxics-1841623 July 2002

Cerium Oxide Enhances the Toxicity of Zinc Oxide Nanoparticles in Human Lung Epithelial Cell Cultures

Rashaideh et al authors.

This study investigated the toxicity of nanomaterials (ZnO) when used in combination with other nanomaterials (CeO2) in the A549 lung epithelial cell line.  Other endpoints investigated were cytokine release and ROS production.  The authors conclude that the presence of CeO2 enhanced the toxicity of ZnO at non toxic levels of CeO2.  This has significance for manufacturers of composite materials which may be incorporated into consumer items and the risks posed to human health.

I have very little to comment on in this paper.

The study is well planned and carried out.  The manuscript is well written, figures clearly presented and the statistical analysis appropriate for the study.

Why did the authors choose CeO2 for the study?  Other so called inert particles such as TiO2 have been used for many years as the standard negative particle control.  However, TiO2 toxicity has been questioned and therefore it would have been worthwhile investigating the role of this particle type in a study of this type.

The authors indicate that further studies would be useful to investigate other endpoints.  I agree with this statement, but I also think it is very important not to base their findings on only one particle combination.  Particles such as CuO and Ag are other particle types which are commonly used and which based on many studies cause toxicity to different cell types in vitro and organ systems in vivo.  The authors should address this limitation of their study in the present submission.

Author Response

Reviewer 1

Reviewer comment

Action/Response

1.       Why did the authors choose CeO2 for the study?  Other so called inert particles such as TiO2 have been used for many years as the standard negative particle control.  However, TiO2 toxicity has been questioned and therefore it would have been worthwhile investigating the role of this particle type in a study of this type.

Investigating the toxicity of TiO2 NPs in combination with ZnO in mammalian cell lines would be a great idea for a future work. We picked CeO2 NPs for several reasons; the material is among the most utilized metal oxides NPs in the industrial sector, it is considered non-toxic, and they are often used in industrial applications together (please see the Introduction section). Furthermore, no regulations were set for the presence of CeO2 NPs in the ambient air of workplaces and any investigation of these NPs in combination with others would shed a light on their potential adverse effects in biological systems.

2.       The authors indicate that further studies would be useful to investigate other endpoints.  I agree with this statement, but I also think it is very important not to base their findings on only one particle combination.  Particles such as CuO and Ag are other particle types which are commonly used and which based on many studies cause toxicity to different cell types in vitro and organ systems in vivo.  The authors should address this limitation of their study in the present submission.

There is innovation in nanomaterials utilizing combinations. Thus, this work evaluates an approach for investigating their effects in vitro. We agree with this suggestion. The following text was added, lines 401-418:

“Undoubtedly, there are other types of NPs that are used in a combination or there might be scenarios where an exposure to a combined effects of several NPs would occur on a wide variety of biological systems. There are several limitation of our study. We have investigated only two types of NPs and their effects on one cell type. However, our previous studies comparing the toxicity of NPs in A549 cells and human primary bronchial epithelial cells (HBEC) (Jing et al., Tox In Vitro, 2015, PMCID: PMC4373347), found similar results between the two cell types for viability, LDH release and IL-8 production. HBECs derived from cadaver lungs have the downside of an unknown exposure history to e.g. cigarette smoke, vaping, or occupational exposures. Further investigation is needed utilizing more advanced biological models and higher number of endpoints. Furthermore, a full physico-chemical characterization of both nanomaterials used in dry form as well as in media, including toxicokinetc and toxicodynamic effects, media and cell dosimetry would have improved an interpretation of our results. However even considering these limitations, our data suggest regulators should take into the account the presence of multiple nanomaterials for health risk assessment.

Reviewer 2 Report

General Comments:

The Rashaideh et al. manuscript reports on an in vitro assessment of mixture toxicity of ZnO and CeO2 nanoparticles to A549 cells using cytotoxicity, LDH, ROS, and chemokine/cytokine release assays based on increased use of materials or technology that use both NPs, thus increasing likelihood of exposure. The authors report that non-toxic CeO2 NPs enhance the ZnO NP toxicity for a majority of the measured endpoints. The study provides new knowledge; however, several major issues were found with model choice, characterization of exposure, and description of methodology that limits and/or questions the findings and conclusions of the study; therefore a decision for publication cannot be reached at this time. Critiques on specific issues are detailed below in the hopes of improving the manuscript.

Major Comments:

1.       Lines 15 and 44. The authors state a study objective that is not clear to the audience since it reads as comparing dose response of two individual NPs.  Please consider revising to include ‘mixture toxicity’. To this reviewer, the main objective of this study was not ‘to compare the toxicity of individual NPs in varying concentrations’ but to assess the toxicity of ZnO and CeO2 NP mixture exposure to single NP exposures. The Introduction nicely introduces the idea of ZnO and CeO2 mixture exposures most likely occur—this thinking should clearly flow into the stated main objective.

2.       Introduction - A one or two sentence overview of known ZnO and CeO2 toxicities in pulmonary exposures would help improve rationale for this study.  ZnO is known to be overtly toxic while CeO2 exposure can cause inflammation and fibrosis in in vivo animal models.

3.       Line 45.  Did the authors have hypothesis to test prior to starting this study?  Please provide.

4.       Particle characterization was partial for the tested CeO2 particles and could be improved, while the authors cite a previous published study that performed a comprehensive characterization of the tested ZnO particle.

5.       Why wasn't each NP characterized in the exposure medium?  Salts and protein will drastically change the agglomerate size, zeta potential, and effective density of each NP, thus directly affecting deposited dose. Furthermore, based on NP and culture medium characteristics, NP tend to agglomerate and settle out of suspension at different rates, thus also impacting the deposited dose on attached cells.  Did the authors consider attempting to quantitatively measure deposited dose of each NP with well-established methods in the literature?  If mass deposition was not 100%, this could have large implications for the interpretation of the presented data.  The authors allude to this fact in the Discussion (Line 301).  The authors should consider providing this data, as it could greatly impact their conclusions.

6.       Line 77. The authors provide no justification for the chosen dose range.  Do these doses reflect ‘environmentally relevant doses’ (Line 283), or another type of exposure? Please provide justification. In addition, why was dosing performed on a mass per volume dose metric, and not a mass per unit area? Inhalation exposures to NPs result in deposition of NP mass on airway and alveolar epithelium which is an area (and not volume)-based exposure.  The authors should provide the administered dose expressed as mass per unit area (ug/cm2) of their 12-well plates.

7.       Why did the authors use a lung adenocarcinoma cell line (A549) to conduct generalized toxicity studies to assess the response of healthy human deep lung epithelial cells?  Little justification was provided. Non-cancerous cells are readily available with published toxicity data with these two NPs to compare to.  The authors briefly acknowledge differences between commercial cell lines and primary cell in their Discussion. Although immortalized or cancer cell lines may provide more consistent results (Line 275), comparative studies have shown them to be more resistant to toxic material.  Primary cells are known to be more sensitive to chemical and particle exposures.  Use of two different cell lines would help assess this issue.

8.       Why were cells exposed to particles suspended in RPMI which differs from cell culture medium DMEM?  Did this change in medium impact overall cell response (was it a confounding factor)?  Did the pH change?  What was the pH for both mediums under culture conditions? Please clarify these issues.

9.       What positive controls were used for the LDH and ROS assays?  A positive control for LDH assay (100% lysis) is used in calculating the % release of LDH from cell cultures, mainly based on the variability of this assay across plates.

9.    Authors report LC50 values, but there is no description of the linear regression method used to calculate these values.  Please supply.

1  .  Authors are to be commended for establishing critical sonication energy for both NPs and measuring the Ce oxidation status; these are two critical pieces of data have not historically been assessed in in vitro toxicity studies.

1.     Figure 3. Please state what each data point represents and the sample size.

1.    Are the observed synergistic mixture effects (Fig. 4 – 8) a product of biological toxicity response, or did the NPs agglomerate differently thus increasing deposited dose to the attached cells (Ie. toxicokinetic or toxicodynamic effect)? Was dissolved ZnO present in cell culture medium during the exposure? The authors suggest this as an explanation for ZnO toxicity (Line 301); why was there no attempt to assess this possibility with ZnO NPs? Further discussion of these ideas in the Discussion with additional data (deposited dose) would increase the findings of this study.

1.    DCF is known to leak out of cells or experience extracellular oxidation under experimental conditions. The method description for this assay is very vague on how the assay was conducted. How confident are the authors that their DCF assay, in fact, measured intracellular ROS? Please improve the method description to increase confidence in these results.

.   . No methods are described for Section 3.5.  Please provide.  Also, placing this Figure and results text is logically out of place since it only gives visual indication of cytotoxicity.  Consider moving it to behind Figure 3 or move to Suppl Material.  Also consider moving the Discussion text associated with Figure 9 to a more logical location.

.    . Line 325-326. The authors attempt to explain reasoning for no IL-1B release into culture supernatants.  Is it possible that activation of IL-1b pathway and cleavage of IL-1b (e.g. NFRP3 inflammasome) did not occur within these cells to illicit IL-1b release? Please give this issue some more thought and adjust the interpretation.

Minor Comments:

1.       Line 246. ‘Level of IL-8’ should read ‘Level of MCP-1’.

2.       Line 331. ‘Figure D’ should read ‘Figure 9’.

3.       Line 362.  Odd phrasing, please revise.

Author Response

Reviewer 2

Comment

Action/Response

1.       Lines 15 and 44. The authors state a study objective that is not clear to the audience since it reads as comparing dose response of two individual NPs.  Please consider revising to include ‘mixture toxicity’. To this reviewer, the main objective of this study was not ‘to compare the toxicity of individual NPs in varying concentrations’ but to assess the toxicity of ZnO and CeO2 NP mixture exposure to single NP exposures. The Introduction nicely introduces the idea of ZnO and CeO2 mixture exposures most likely occur—this thinking should clearly flow into the stated main objective.

Thank you for noting that. We changed that to be: “The aim of this study was to assess the adverse biological outcomes of CeO2-ZnO nano-mixture in human lung epithelial cells.”

2.       Introduction - A one or two sentence overview of known ZnO and CeO2 toxicities in pulmonary exposures would help improve rationale for this study.  ZnO is known to be overtly toxic while CeO2 exposure can cause inflammation and fibrosis in in vivo animal models.

We added more details regarding the toxicity of ZnO NPs , but did not focus as much on the toxicity of CeO2 since it is  covered more fully in the discussion section.

3.       Line 45.  Did the authors have hypothesis to test prior to starting this study?  Please provide.

Our a priori hypothesis was that the presence of CeO2 NPs with ZnO NPs will augment the adverse biological outcomes in human lung epithelial cells. Our hypothesis has been added to the Introduction (lines 64-66).

4.       Particle characterization was partial for the tested CeO2 particles and could be improved, while the authors cite a previous published study that performed a comprehensive characterization of the tested ZnO particle.

We acknowledge that there was less characterization of the CeO2 in the discussion. We have added this to limitations of our study (lines 401-418, also see above).

5.       Why wasn't each NP characterized in the exposure medium?  Salts and protein will drastically change the agglomerate size, zeta potential, and effective density of each NP, thus directly affecting deposited dose. Furthermore, based on NP and culture medium characteristics, NP tend to agglomerate and settle out of suspension at different rates, thus also impacting the deposited dose on attached cells.  Did the authors consider attempting to quantitatively measure deposited dose of each NP with well-established methods in the literature?  If mass deposition was not 100%, this could have large implications for the interpretation of the presented data.  The authors allude to this fact in the Discussion (Line 301).  The authors should consider providing this data, as it could greatly impact their conclusions.

We agree with the reviewer, and we have included this into the limitations for our study, see above.

6.       Line 77. The authors provide no justification for the chosen dose range.  Do these doses reflect ‘environmentally relevant doses’ (Line 283), or another type of exposure? Please provide justification. In addition, why was dosing performed on a mass per volume dose metric, and not a mass per unit area? Inhalation exposures to NPs result in deposition of NP mass on airway and alveolar epithelium which is an area (and not volume)-based exposure.  The authors should provide the administered dose expressed as mass per unit area (ug/cm2) of their 12-well plates.

We have stated that the NP exposure doses were selected based on our previous studies (lines 69-70). We have also expressed the doses used per surface area. The cell growth surface area of 12-well plate is 3.8 cm2, 1 mL of each NP was used per each well. Following text was added, lines 109-110:

“These concentrations correspond to 0.5, 1, 2, 4, 8 and 16 µg/cm2 per each NP per each well.”

7.       Why did the authors use a lung adenocarcinoma cell line (A549) to conduct generalized toxicity studies to assess the response of healthy human deep lung epithelial cells?  Little justification was provided. Non-cancerous cells are readily available with published toxicity data with these two NPs to compare to.  The authors briefly acknowledge differences between commercial cell lines and primary cell in their Discussion. Although immortalized or cancer cell lines may provide more consistent results (Line 275), comparative studies have shown them to be more resistant to toxic material.  Primary cells are known to be more sensitive to chemical and particle exposures.  Use of two different cell lines would help assess this issue.

The A549 cells have been widely used to model the alveolar Type II pulmonary epithelium. A549 is the only commercially available cell line that represents this region of the lung and has the characteristics of alveolar type II cells. We previously published a comparison of the toxicity of nanoparticles to A549 cells and human primary bronchial epithelial cells (HBEC) (Jing et al., Tox In Vitro, 2015, PMCID: PMC4373347). We found similar results between the two cell types for viability, LDH release and IL-8 production.  HBECs derived from cadaver lungs have the downside of an unknown exposure history to e.g. cigarette smoke, vaping, or occupational exposures. We have included a discussion of this in the manuscript, lines 408-416.

8.       Why were cells exposed to particles suspended in RPMI which differs from cell culture medium DMEM?  Did this change in medium impact overall cell response (was it a confounding factor)?  Did the pH change?  What was the pH for both mediums under culture conditions? Please clarify these issues.

DMEM is recommended for adherent cells like A549 cells. While RPMI is recommended for suspended cells. We grew those cells before in 4 supplemented media: DMEM; EMEM; RPMI; and KF12. We did not notice any difference in the doubling time or in the morphology of the cells under the microscope. Initially we were culturing and splitting the cells with DMEM as recommended by the supplier. While we used RPMI in the treatments since most of the studies used this medium for submerged nanotoxicology studies. Controls were maintained in particle free DMEM media and no effect was noticed when switching to RPMI at the treatment time of other cells.

9.       What positive controls were used for the LDH and ROS assays?  A positive control for LDH assay (100% lysis) is used in calculating the % release of LDH from cell cultures, mainly based on the variability of this assay across plates. 

L-LDH from rabbit muscles was used to create a standard curve as recommended by the manufacturer for the LDH assay. The manufacturer recommends having a background control, low control, and high control. We had the background control and the low control (untreated cells). Any results out of the standard range were excluded. While some studies use H2O2 as a positive control for ROS, the protocol that we used did not include a peroxide control.

10.   Authors report LC50 values, but there is no description of the linear regression method used to calculate these values. Please supply.

Half-maximal inhibitory concentration (IC50) is calculated by plotting the increasing concentrations of ZnO vs. the viability of A549 cells (dose-response data).

Y = a * X + b,

IC50 = (0.5 - b)/a.

The x-axis (dose) is logarithm-transformed.

11.   Authors are to be commended for establishing critical sonication energy for both NPs and measuring the Ce oxidation status; these are two critical pieces of data have not historically been assessed in in vitro toxicity studies.

Thank you.

12.   Figure 3. Please state what each data point represents and the sample size.

The points represent the increasing concentrations of ZnO vs. the viability of A549 cells (viability assay). As mentioned above the x-axis (dose) was logarithm-transformed. N=8 

13.   Are the observed synergistic mixture effects (Fig. 4 – 8) a product of biological toxicity response, or did the NPs agglomerate differently thus increasing deposited dose to the attached cells (Ie. toxicokinetic or toxicodynamic effect)? Was dissolved ZnO present in cell culture medium during the exposure? The authors suggest this as an explanation for ZnO toxicity (Line 301); why was there no attempt to assess this possibility with ZnO NPs? Further discussion of these ideas in the Discussion with additional data (deposited dose) would increase the findings of this study.

14.   DCF is known to leak out of cells or experience extracellular oxidation under experimental conditions. The method description for this assay is very vague on how the assay was conducted. How confident are the authors that their DCF assay, in fact, measured intracellular ROS? Please improve the method description to increase confidence in these results.

To examine the intracellular ROS in our cells, we used the carboxy derivative of fluorescein (carboxy-H2DCFDA (C400)). The carboxy derivative carries extra negative charges that improve its retention with the cell compared to non-carboxylated forms. The nonfluorescent probe works when undergoes intracellular hydrolysis by esterase. To minimize the possibility of any extracellular hydrolysis, cells were washed with PBS as recommended by the manufacturer. Then we lysed the cells with 0.1% Triton X-100 in PBS. We have modified the methodology section to include this additional information lines 145-152.

“After the treatment, cells were washed with phosphate buffer saline (PBS). The exposure medium was collected for further assays. Hanks Balanced Salt Solution (HBSS) containing carboxy-H2DCFDA (25 µM, final concentration) was added to the cells for 30 min incubation at 37°C under dark condition. Following incubation, HBSS solution was removed and cells were washed twice with 200 uL of PBS and were then lysed with 400 µL of 0.1% Tri-ton X-100 in PBS for 30 min under dark condition. After lysis, the suspensions were centrifugated at 14,000 x g for 15 min at 4°C and the supernatant was taken and placed in the 96-well plate for measurement.”

15.   No methods are described for Section 3.5.  Please provide.  Also, placing this Figure and results text is logically out of place since it only gives visual indication of cytotoxicity.  Consider moving it to behind Figure 3 or move to Suppl Material.  Also consider moving the Discussion text associated with Figure 9 to a more logical location.

We agree with this suggestion and have moved the microscopic images to the supplement.

16.   Line 325-326. The authors attempt to explain reasoning for no IL-1B release into culture supernatants.  Is it possible that activation of IL-1b pathway and cleavage of IL-1b (e.g. NFRP3 inflammasome) did not occur within these cells to illicit IL-1b release? Please give this issue some more thought and adjust the interpretation.

This excellent suggestion was added to the discussion section, line 370-373.

“In the present study, IL-1b was not detectible in either the treated or control cells. It may be that activation of the IL-1b pathway and cleavage of IL-1b did not occur within these cells to illicit IL-1b release. This finding could also be explained by the mediator’s short lifetime [37-39].

Minor Comments:

1.    Line 246. ‘Level of IL-8’ should read ‘Level of MCP-1’.

2.       Line 331. ‘Figure D’ should read ‘Figure 9’.

3.       Line 362.  Odd phrasing, please revise.

·      Now this is Figure 7. The legend was corrected (now line 269).

·      This figure has been moved to the supplement as recommended by reviewers (Figure is now S3, supplementary)

·      Done. (Now line 404)

Reviewer 3 Report

 In this manuscript, the authors evaluated ZnO or CeO2 NPs alone or as a mixture induced cell viability, reactive oxygen species (ROS) formation, cell membrane integrity, and cytokine production. A series of experiments were carried out to demonstrate that the presence of CeO2 enhanced the toxicity of ZnO in A549 cells. There are some questions needed to be addressed.

1.    There are too many errors to modify. Such as line 26: “This suggests additive toxicity of these two nanomaterials.”; line163: “3.1. Viability Assay”; line266: “Studies have used various models to evaluate the toxicity of monotypic nanoparticles.” Where is figure 2? The authors should rewrite the introduction and discussion sections

2.    Line 273 : Historically, most in vitro studies have utilized commercial cell lines rather than primary cells…. What does the author want to explain? The first paragraph in the discussion section is strange to me.

3.    This result “3.5. Microscopic Images” should delete. Some of these information can add in 3.2 viability assay.

4.    Why were 2.0, 3.9, 7.8, 15.6, 31.3, and 62.5 µg/mL NPs chosen for in vitro study? It is very high dose.

Author Response

Reviewer 3

There are too many errors to modify. Such as line 26: “This suggests additive toxicity of these two nanomaterials.”; line163: “3.1. Viability Assay”; line266: “Studies have used various models to evaluate the toxicity of monotypic nanoparticles.” Where is figure 2? The authors should rewrite the introduction and discussion sections.

Substantial changes and improvements were made as suggested by Reviewers 1 and 2.

Line 273 : Historically, most in vitro studies have utilized commercial cell lines rather than primary cells…. What does the author want to explain? The first paragraph in the discussion section is strange to me.

The statement was justifying the usage of commercial cell lines instead of primary cells for cell-based studies. See the response to Reviewer 2 item 7.

T   This result “3.5. Microscopic Images” should delete. Some of these information can add in 3.2 viability assay.

Why were 2.0, 3.9, 7.8, 15.6, 31.3, and 62.5 µg/mL NPs chosen for in vitro study? It is very high dose.

Per Reviewer 2, item 15, we have moved the microscopic images to the supplement to support the results of the viability assay.

10, 25, 50, 100 and 200 μg/mL are the main doses (concentrations) used for in vitro nano-toxicity studies in order to observe dose-response relationships. In this study, we did 2-fold dilution starting from a stock of 500 μg/mL until we reached the doses that were used in our study. Our doses are within the common range (10 -200 μg/mL) used in many published studies. These concentrations correspond to 0.5, 1, 2, 4, 8 and 16 µg/cm2 per well.

Round 2

Reviewer 2 Report

The authors made considerable effort to improve their manuscript and addressed a majority of the previous issues raised.

Several issues remain unaddressed or can be considered missed opportunities in the manuscript:

1.       It is still not clear why the authors did not attempt to quantitively measure deposited dose using the published distorted grid methods of Deloid etal. 2017, which several of the authors clearly used in their previous publication in 2020 for 9 different ENMs, including ZnO and CeO2 particles (doi: 10.1016/j.impact.2020.100214). The authors’ rebutall states that they are in agreement with this view. The absence of this critical piece of data raises some doubt on the accuracy of the conclusions stated in this manuscript and is considered a missed opportunity to strengthen the manuscript.  This limitation is briefly acknowledged in the Discussion. The authors are encouraged to evaluate deposited dose in their future studies.

2.       Please provide a statement in the Methods Section stating that switching between DMEM and RPMI for NP exposures appeared to not alter cell morphology or doubling times.

3.       The absence of using positive controls in the LDH and ROS assays is worrisome, more so for the LDH assay. The high control helps bracket the cellular response since the assay varies from plate to plate.  The authors should take note to use all three LDH controls in future studies.

4.       Please state the methodology used to calculate IC50 values within the Method text.  This was stated in the rebuttal to the reviewers but did not make its way into the manuscript.

5.       Are the observed synergistic mixture effects (Fig. 4 – 8) a product of biological toxicity response, or did the NPs agglomerate differently thus increasing deposited dose to the attached cells? A brief discussion of this could help future investigations into NP mixtures. This was not addressed in the revision.

6.       It was not clear if the citations included in the revised text made it into the References section—please doublecheck.

Author Response

Toxics-1841623-2

R2 Response to Reviewer 2

Comment

Action/Response

1. It is still not clear why the authors did not attempt to quantitively measure deposited dose using the published distorted grid methods of Deloid etal. 2017, which several of the authors clearly used in their previous publication in 2020 for 9 different ENMs, including ZnO and CeO2 particles (doi: 10.1016/j.impact.2020.100214). The authors’ rebuttal states that they are in agreement with this view. The absence of this critical piece of data raises some doubt on the accuracy of the conclusions stated in this manuscript and is considered a missed opportunity to strengthen the manuscript.  This limitation is briefly acknowledged in the Discussion. The authors are encouraged to evaluate deposited dose in their future studies.

We appreciate the additional comments from Reviewer 2 – thank you.

We understand that uncertainty of dosimetry is a limitation of this work and one we plan to address going forward. The distorted grid (DG) computational model is designed to predict the dosimetry of certain nanosuspensions, such as polydisperse high-aspect ratio suspensions and allows dissolution modeling for soluble or partially soluble ENMs in individual form. Unfortunately, we did not perform the needed measurements under the conditions of use because we were under the impression that the DG model was not validated for NP mixtures such as ours. Yesterday, Dr. Thorne contacted Dr. Demokritou, the senior author of that paper, to ask about application of this method to mixtures of nanoparticles with differing characteristics. He indicated that “the methodology can be applied to any particles inclusive of real-world environmental nanoparticles and mixtures as long as this is a mixture of NPs and doesn’t contain some large fine particles.” ”…the characterization of the average effective density of particle agglomerates needs to be measured…".

We looked into our archives from our prior work (Adamcakova-Dodd et al. Part Fiber Tox; 2014; Areecheewakul et al. NanoImpact, 2020) with ZnO and CeO2 and while we have appropriate characterization data for both metal oxides, we do not have it for the same nanosized materials from the same supplier as we used in this study. These nanomaterials are approximately half the primary size as those used in our earlier work. In addition, the ZnO and CeO2 NP were in a different medium in the Areecheewakul NanoImpact publication. Thus, it is most prudent to acknowledge we did not apply the DG model and state that as a limitation. As we do further work with mixtures we will explore the DG model more fully and evaluate it’s applicability in our system. To that end, we have added this paragraph to the discussion:

“There are several limitations of our study. We have investigated only two types of NPs and their effects on one cell type. However, our previous studies comparing the toxicity of NPs in A549 cells and human primary bronchial epithelial cells (HBEC) [47], found similar results between the two cell types for viability, LDH release and IL-8 production. HBECs derived from cadaver lungs have the downside of an unknown exposure history to e.g., cigarette smoke, vaping, or occupational exposures. The uncertainty of cell dosimetry is another limitation of this study. The distorted grid (DG) computational model has been suggested to predict the dosimetry of certain nanosuspensions [17]. This model was not applied in this study due to the absence of characterization parameters, such as the average effective density of particle agglomerates, for the 36 different individual and mixture concentrations studied. Our future work will more fully characterize the mixtures and attempt to validate the DG model for NP combinations utilizing more advanced biological models and an expanded list of endpoints. Lastly, a more expansive physicochemical characterization of the nanomaterials in media would have aided interpretation of our results.”

2. Please provide a statement in the Methods Section stating that switching between DMEM and RPMI for NP exposures appeared to not alter cell morphology or doubling times.

The statement was added (line 113-117).

“Switching from DMEM to 1640 RPMI did not affect the doubling time or the morphology of the cells. Initially, we were culturing and splitting the cells with DMEM as recommend-ed by the supplier. We used 1640 RPMI in the treatments since most researchers have used this medium for submerged nanotoxicology studies.”

3. The absence of using positive controls in the LDH and ROS assays is worrisome, more so for the LDH assay. The high control helps bracket the cellular response since the assay varies from plate to plate.  The authors should take note to use all three LDH controls in future studies.

For LDH, the manufacturer recommended (high and low controls) but not (positive and negative) controls. The manufacturer elucidated that two methods could be used; one of them required a standard curve (separate from the kit). Our results were always within the range of the standard curve. Any out-of-range points were excluded. For ROS, positive control like hydrogen peroxide is usually recommended to be used for ROS assays. The protocol we had in our lab at the time of this work did not use one.

4. Please state the methodology used to calculate IC50 values within the Method text.  This was stated in the rebuttal to the reviewers but did not make its way into the manuscript.

The statement was added (line 129-132).

“The half-maximal inhibitory concentration (IC50) was calculated by plotting the increasing concentrations of ZnO vs. the viability of A549 cells (dose-response data).

Y = a * X + b, IC50 = (0.5 - b)/a.

The x-axis (dose) was logarithm-transformed.”

5. Are the observed synergistic mixture effects (Fig. 4 – 8) a product of biological toxicity response, or did the NPs agglomerate differently thus increasing deposited dose to the attached cells? A brief discussion of this could help future investigations into NP mixtures. This was not addressed in the revision.

The agglomerates might play a role in the toxicity that we have seen. Many studies reported that large and dense agglomerates caused more toxic effects than smaller and lighter agglomerates. We do not know how much media those agglomerates contain since we do not know the effective density of the CeO2-ZnO mixture in the medium. The amount of medium affects the toxicity of the agglomerate too. The presence of ZnO along with CeO2 could affect the sedimentation rate of CeO2 too.

6. It was not clear if the citations included in the revised text made it into the References section—please doublecheck.

The references were updated. See Lines: 66, 329, 335, and 337.

Reviewer 3 Report

All my comments had been revised.

Author Response

Reviewer 3 states that "All my comments had been revised." Thank you.